# Sex-Specific Relationships of Physical Activity and Sedentary Behaviour with Oxidative Stress and Inflammatory Markers in Young Adults

**DOI:** 10.3390/ijerph20020899

**Published:** 2023-01-04

**Authors:** Juan Corral-Pérez, Martin Alcala, Daniel Velázquez-Díaz, Alejandro Perez-Bey, María Á. Vázquez-Sánchez, Maria Calderon-Dominguez, Cristina Casals, Jesús G. Ponce-González

**Affiliations:** 1ExPhy Research Group, Department of Physical Education, Faculty of Education Sciences, University of Cadiz, 11519 Cadiz, Spain; 2Biomedical Research and Innovation Institute of Cadiz (INiBICA) Research Unit, Puerta del Mar University Hospital, 11009 Cadiz, Spain; 3Department of Chemistry and Biochemistry, Facultad de Farmacia, Universidad San Pablo-CEU, CEU Universities, 28003 Madrid, Spain; 4Advent Health Research Institute, Neuroscience Institute, Orlando, FL 32803, USA; 5GALENO Research Group, Department of Physical Education, Faculty of Education Sciences, University of Cadiz, 11519 Cadiz, Spain; 6Department of Nursing, Faculty of Health Sciences, PASOS Research Group and UMA REDIAS Network of Law and Artificial Intelligence applied to Health and Biotechnology, University of Malaga, 29071 Malaga, Spain; 7Biomedicine, Biotechnology and Public Health Department, University of Cadiz, 11002 Cadiz, Spain

**Keywords:** accelerometry, antioxidants, inflammation, sedentarism, gender

## Abstract

This study aims to analyse sex-specific associations of physical activity and sedentary behaviour with oxidative stress and inflammatory markers in a young-adult population. Sixty participants (21 women, 22.63 ± 4.62 years old) wore a hip accelerometer for 7 consecutive days to estimate their physical activity and sedentarism. Oxidative stress (catalase, superoxide dismutase, glutathione peroxidase, glutathione, malondialdehyde, and advanced oxidation protein products) and inflammatory (tumour necrosis factor-alpha and interleukin-6) markers were measured. Student *t*-tests and single linear regressions were applied. The women presented higher catalase activity and glutathione concentrations, and lower levels of advanced protein-oxidation products, tumour necrosis factor-alpha, and interleukin-6 than the men (*p* < 0.05). In the men, longer sedentary time was associated with lower catalase activity (*β* = −0.315, *p* = 0.04), and longer sedentary breaks and higher physical-activity expenditures were associated with malondialdehyde (*β* = −0.308, *p* = 0.04). Vigorous physical activity was related to inflammatory markers in the women (tumour necrosis factor-alpha, *β* = 0.437, *p* = 0.02) and men (interleukin−6, *β* = 0.528, *p* < 0.01). In conclusion, the women presented a better redox and inflammatory status than the men; however, oxidative-stress markers were associated with physical activity and sedentary behaviours only in the men. In light of this, women could have better protection against the deleterious effect of sedentarism but a worse adaptation to daily physical activity.

## 1. Introduction

Oxidative stress is defined as an imbalance between oxidants and antioxidants in favour of oxidants, leading to a disruption of redox signalling and control or molecular damage [1]. This imbalance, with an overproduction of reactive oxygen species (ROS), leads to oxidative damage to different macromolecules, such as lipids, proteins, and DNA, resulting in cell death through the necrotic and apoptotic processes [2]. Moreover, oxidative stress increases the level of proinflammatory cytokines, such as interleukin−6 (IL−6) and tumour necrosis factor-alpha (TNF-α) [3].

However, ROS and inflammatory cytokines, produced in limited quantities, are also involved in the regulation of cell homeostasis and other functions, such as signal transduction, gene expression, and the activation of receptors [4]. Although physical exercise produces an acute elevation of ROS and proinflammatory cytokines of skeletal-muscle origin, the basal plasma levels of these markers are lowered with regular physical exercise [5,6], highlighting their role in the signalling pathways regulating muscle adaptation to exercise [7].

Sex-specific oxidative stress and inflammatory responses to exercise and training have been poorly studied [8]. Previous studies suggested that women present higher antioxidant capacity and lower levels of oxidant markers compared to men at rest [9,10], and, accordingly, differences between sexes in muscle mitochondrial function have also been reported [11]. Furthermore, levels of physical activity and sedentary time may differ depending on sex [12]; specifically, women tend to perform physical activities with lower intensity but also spend less time performing sedentary activities than men [13]. However, most studies evaluating the associations between physical activity and oxidative stress have been performed on older populations [14], not on young adults, in whom physiological differences between sexes are highlighted. Thus, research on the sex-specific associations of daily physical activity and sedentary behaviour with oxidative stress and the inflammatory status of a young adult population may contribute to a better understanding of the different adaptations to training found between males and females [15].

For all these reasons, the present study aimed to evaluate the associations between daily physical activity and sedentary behaviour with oxidative stress and inflammatory markers according to sex. 

## 2. Materials and Methods

### 2.1. Design

This study is a cross-sectional design of the “NutAF research Project”, which aimed to investigate the relationships between nutritional habits and the level of physical activity with appetite-regulating hormones and genetic polymorphisms related to obesity in young adults [16,17,18,19]. The study was carried out following the Helsinki Declaration and was approved by the Hospital Puerta del Mar Ethical Committee (Cadiz, Spain). The participants were fully informed of the aims of the study and any possible side effects; all participants gave their written informed consent before measurements. 

The physical-activity levels and sedentary behaviour were estimated using accelerometers for seven consecutive days immediately before the day of blood-sample collection. Participants were asked to maintain normal daily physical activity during the week, except for the day before measurements, on which they were instructed to avoid vigorous physical exercise. Furthermore, the morning of blood sampling, the participants avoided active transport (walking, cycling, etc.). After returning the accelerometer, fasting blood samples were taken from the antecubital vein, collected in EDTA tubes, and centrifuged at 2500 rpm, for 15 min, at 4 °C to obtain plasma and red blood cells, which were aliquoted and stored at −80 °C until further analyses. 

### 2.2. Participants

A total of 60 young Spanish adults (21 women) were recruited. Participant recruitment consisted of mailed flyers describing the research and contact information; the flyers were also published on social media (Facebook, Instagram, etc.). Printed flyers were posted on academic campus (at bus stops, housing complexes, and near dorms) and around the community, targeting areas with high traffic of young adults. Inclusion criterion was age between 18 and 35 years. Exclusion criteria included: (*i*) change to usual diet in the 6 preceding months, (*ii*) being an active smoker, (*iii*) having any cardiovascular disease or any condition that could prevent physical activity, (*iv)* vitamin supplementation in the 6 preceding months, and (*v)* having a macronutrient or micronutrient deficit in the usual diet.

Before the evaluation, participants completed a dietary record for five consecutive days, including two weekend days. Energy, macronutrient, and micronutrient intakes were evaluated using DIAL software (version 1.19) to detect any deficits.

### 2.3. Physical Activity and Sedentary Behaviour

The triaxial ActiGraph GT3X+ accelerometer (ActiGraph, Pensacola, FL, USA) was attached tightly to the hip, on the back side, during waking hours except for water-based activities, according to established procedures [20], registering during 7 consecutive days. This position of the accelerometer allowed us to accurately estimate physical activity and sedentary behaviour [21]. 

The measurement rate was set at 100 Hz and raw data were downloaded in epoch lengths of 60s [22]. Only the participants wearing the accelerometers for ≥10 h/day during at least 3 weekdays and 1 weekend day were included in the final analyses [22]. Choi’s algorithm with default settings was used for wear-time validation [23]. Participants were asked to remove the accelerometer during hours of sleep. 

Accelerometer data were analysed using ActiLife 6.6.2 software (ActiGraph, Florida, USA). The Freedson Adult VM3 (2011) cut points were utilized to classify physical activity into light (LPA, 150–2689 counts/min), moderate (MPA, 2690–6166 counts/min), vigorous (VPA, 6167–9642 counts/min), and very vigorous (VVPA, ≥9643 counts/min) [24]. Moderate-to-vigorous physical activity (MVPA) was considered any activity from moderate to very vigorous (i.e.， ≥2690 counts/min). Additionally, activity energy expenditure was estimated using the Freedson VM3 (2011) equation. Sedentary behaviour (classified as <150 counts/min), sedentary bouts as the time accumulated in a consecutive period of >10 min, the mean number of sedentary bouts, and the time spent in sedentary bouts were calculated per day. Furthermore, sedentary breaks were considered whenever participants were above 150 counts/min after a sedentary bout, and the time, number, and length of sedentary breaks were calculated per day [25].

### 2.4. Oxidative Stress and Inflammatory Markers

Hemolysate was used for the measurement of antioxidant-enzyme activity and the non-enzymatic antioxidant, reduced glutathione (GSH). The catalase (CAT)-specific activity was measured by monitoring the disappearance of hydrogen peroxide at 240 nm. To measure superoxide dismutase (SOD) enzymatic activity, we adapted the method described by Bamforth [26]. It is based on the xanthine/xanthine-oxidase reaction for superoxide anion generation (O_2_^−^) and reduction of cytochrome c for detection at 546 nm. The glutathione peroxidase (GPx)-specific activity assay was based on the oxidation of GSH by GPx. Reduced glutathione is regenerated by glutathione reductase, using NADPH  +  H^+^ as a cofactor. The reaction rate was measured following the disappearance of NADPH  +  H^+^ at 340 nm. The amount of GSH was fluorometrically detected in the presence of o-phtaldialdehyde (OPD). First, 50 µl of sample were deproteinized with 150 µl of 5 % trichloroacetic acid to prevent non-specific crossed reactions between the OPD and the cysteine residues. The supernatant was then neutralized with 0.1 M NaOH and incubated with 10 µl of 1 mg/mL OPD. Fluorescent products were measured at Ex = 350 nm/ Em = 420 nm and the results were interpolated in a GSH-calibration curve.

Plasma was used for the quantification of oxidative-damage markers. Malondialdehyde (MDA) was measured by measuring the final products of lipid peroxidation with thiobarbituric acid at 532 and 553 nm [27]. Advanced oxidation protein products (AOPPs) were measured following the transformation of iodide to diatomic iodine under acidic conditions; this reaction can be spectrophotometrically followed at 340 nm [28]. Plasma IL−6 and TNF-α levels were determined by the MILLIPLEX^®^ MAP Human High Sensitivity T Cell Magnetic Bead Panel 96-Well Plate Assay (HSTCMAG−28SK, Merck Millipore) and the Luminex^®^ 200TM System (Luminex Corp., Austin, TX, USA) according to the manufacturer’s instructions. The intra-assay coefficients of variation were <5 %, and the inter-assay coefficients of variation were <15 %. The minimum detected value for IL−6 was 0.11 pg/mL, and the value for TNF-α was 0.16 pg/mL.

### 2.5. Statistical Analysis

All data are expressed as mean ± standard deviation (SD). The Kolmogorov–Smirnov and Levene tests were used to check the normality of distribution and homogeneity of variance, respectively. An independent Student t-test sample was obtained to analyse the differences between sexes in all outcomes. Interaction analyses showed a significant interaction of sex in most of the variables; therefore, the analyses were conducted separately by sex. Single linear regressions were performed to examine the associations between physical activity and sedentary behaviour with oxidative stress and inflammatory markers. All analyses were performed using the IBM SPSS Statistics 22 software (SPSS Inc., Chicago, IL, USA), with significance set at *p* < 0.05.

## 3. Results

The participants wore their accelerometers for 6.8 ± 0.4 days. Regarding the levels of physical activity, international recommendations establish a minimum of >150 min per week of MVPA. In our analyses, almost all the participants met this recommendation (98.33 %). The participants’ characteristics by sex are presented in Table 1, providing data on sedentary behaviour, physical activity, oxidative stress, and inflammatory markers. There were no significant differences in sedentary behaviour and physical activity between the sexes; nevertheless, the women showed significantly higher levels of CAT and GSH activity and lower levels of AOPPs, IL−6, and TNFα than the men.

Significant associations between accelerometer outcomes and oxidative stress and inflammatory markers were found in this study. Table 2 shows the associations between sedentary and physical-activity behaviour and antioxidant enzymes. In the men only, a longer sedentary time was inversely associated with CAT activity, and physical-activity expenditure was inversely associated with SOD activity. Table 3 presents the associations of the accelerometer outcomes with the pro-oxidants and inflammatory markers in the men and women. Maximal time in sedentary breaks and physical activity expenditure were also negatively associated with MDA in the men only. Finally, regarding the inflammatory markers, higher daily VPA and VVPA were associated with higher resting levels of TNF-α in the women and IL−6 in the men.

## 4. Discussion

This study aimed to compare sex-specific differences in resting oxidative stress and inflammatory markers in young adults and to analyse the possible relationships of these markers with sedentary and physical activity behaviours. We showed that the women had lower oxidative stress at rest with higher antioxidant activities and lower concentrations of oxidant and inflammatory markers than the men. Additionally, longer periods of sedentary behaviour were associated with decreased CAT activity only in the men, suggesting that women are better protected against the deleterious effect of sedentarism. Finally, higher daily VPA and VVPA were associated with elevated resting levels of TNF-α and IL−6 in the women and men, respectively, which may be related to the high levels performed by our participants during the week.

Hence, our data showed that the women had raised antioxidant activity and lower oxidative stress and inflammation than the men, with increased CAT activity and GSH concentration and decreased levels of AOPPS, TNF-α, and IL−6, suggesting that women are better protected against oxidative stress and inflammation [10,29]. This enhanced defence may be due to the lower levels of peroxide production that have been shown by female compared to male mitochondria, with female rats producing half of the amount of H_2_O_2_ compared to male rats [30]. This reduction in the production of pro-oxidants has been attributed to the protective effect of oestrogen, a sex-specific hormone which has a double effect on oxidative stress. Oestrogen could act as an antioxidant by scavenging ROS [31], as well as upregulating the expression of antioxidant enzymes [30], explaining why the women showed higher antioxidant enzymes than the men in our study. This increased defence against oxidative damage could explain why women usually have a longer lifespan [32].

Regarding sedentary behaviour, a significant association between sedentary time and CAT activity was found in men, meaning that the men who spent more time engaging in sedentary behaviour also showed lower CAT activities, but this did not happen in the women. In line with this observation, other authors have found higher antioxidant capacity in active young adults compared to sedentary young adults [33]. Similarly, this study showed that the males who spent longer times in sedentary breaks also had reduced levels of MDA, whereas this association was not found in the women. To the best of our knowledge, this is the first study to have found a sex-specific association between sedentary breaks and oxidative stress.

The accumulation of longer sedentary periods may lead to chronically stressed mitochondria and decreased antioxidant-enzyme activities [34]. These antioxidant enzymes are the primary defence against the deleterious effects of ROS and decreased activity eventually leads to increased oxidative stress, especially in men whose critical molecules are more prone to suffering from oxidative damage [35]. Thus, breaking with sedentary behaviours through longer sedentary breaks may be an efficient strategy to improve the redox status in young adult men by decreasing the levels of oxidative damage, as suggested by our results, which would result in a reduced risk of suffering from cardiometabolic disease [36]. The lack of significant associations in women may be related to the increased activity and expression of females' mitochondrial antioxidant enzymes [35], which could act as a protective factor against the harmful effects of the accumulation of longer sedentary periods. 

In line with these results, higher levels of physical activity were significantly associated with lower levels of MDA in the men only. Higher levels of physical activity are associated with mitochondrial biogenesis and antioxidant defence [37,38], which may explain the reduced levels of MDA in males. Higher levels of physical activity may be related to decreased sedentary time, supporting the idea that men are more vulnerable to the damaging impact of sustained oxidative stress induced by sedentarism. The absence of a significant association in our female population might be related to the lower ROS levels produced by female mitochondria, which prevent exercise-induced adaptions, preventing health benefits; thus, males could obtain more health benefits due to the higher level of exercise-induced oxidative stress and the higher ROS exposure of the mitochondria [39]. 

The regular practice of physical activity has been associated with an increased antioxidant capacity and a reduction of the concentration of pro-oxidants in adults [40]. Nonetheless, in our study, no significant associations were found between any specific physical activity behaviour and oxidative-stress markers in both males and females. This lack of associations could be explained by the high levels of physical activity our participants performed weekly. Almost 95 % of our participants met the physical activity recommendations of the World Health Organization for adults of >150 min/week of MPA or >75 min/week of VPA or an equivalent combination of MVPA throughout the week [41]. Consequently, it is difficult to find significant benefits of physical activity for oxidative-stress markers when the recommendations are met. 

Regarding inflammatory status, little is known about the associations between the intensity of accelerometer-measured physical activity and inflammatory markers in adults [42]. In this study, we found significant sex-specific associations of different vigorous physical levels with inflammatory markers. In the men, higher levels of VVPA were significantly associated with increased levels of IL−6, while in the women, higher levels of VPA were associated with elevated levels of TNF-α. It has been shown that high-intensity training can produce irritation in the exercised muscles, which increases inflammatory markers such as TNF-α and IL−6 locally, after which they are poured into blood plasma [43]. Nevertheless, regular physical exercise can modulate these inflammatory markers by lowering their basal levels; hence, the intensity of the physical activity performed by our subjects could have been excessive for their levels of physical fitness. Additionally, the sex-specific associations may be explained by the differences in sex hormones. Testosterone has been shown to inhibit the production and secretion of TNF-α, contrary to estradiol, which may explain why this association is only found in women [44]. Regarding IL−6, although it has been stated that women have a delayed response to IL−6 levels [45], exercise increases the cytokine expression in men [46], which may explain why the association only appears in male participants. 

This study has several limitations. Firstly, since it is a cross-sectional study, causal relationships cannot be identified. Secondly, studies with a larger sample size, particularly for women, whose sample was small in the present study, are encouraged. Finally, it is important to highlight the low strength of the relationships reported in most of the significant associations (low R-squared) indicating a high variability, since other relevant factors could influence these outcomes. However, several biochemical parameters strengthened the study. Moreover, both physical activity and sedentary behaviour were measured using accelerometry, a more objective method than self-reported questionnaires. Future studies with an experimental design are encouraged in order to determine the impact of different exercise programs on oxidative stress and inflammatory markers according to sex, ideally including not only blood samples but also muscle biopsies.

## 5. Conclusions

We found that the women had lower levels of oxidative stress and inflammatory markers at rest than the men, with increased CAT activity and GSH concentration, and decreased levels of AOPPS, TNF-α, and IL−6 in the female young adults than in the males. Additionally, sex-specific associations were found between physical activity and sedentary behaviour and oxidative stress and inflammation. Specifically, longer sedentary periods were associated with lower CAT activity in the men only; therefore, breaking that sedentary behaviour may be an optimal strategy to maintain a good redox status in young adult men. These results suggest that women are better protected against the stressors of sedentarism than men. Regarding inflammation, although physical activity regulates the release and activity of IL−6 and TNF-α, our results showed that the young adults who performed physical activity of great intensity (VPA and VVPA) presented increased IL−6 and TNF-α levels.

## Figures and Tables

**Table 1 ijerph-20-00899-t001:** Participant characteristics by sex.

	Total (*n* = 60)			Men (*n* = 39)			Women (*n* = 21)			*p*
General characteristics										
Age (years)	22.63	±	4.62	22.41	±	3.84	23.05	±	5.88	0.614
Height (cm)	171.72	±	8.70	176.17	±	6.14	163.46	±	6.44	<0.001
Body mass (kg)	74.55	±	14.75	76.99	±	13.01	70.02	±	16.93	0.081
Body-mass index (kg/m^2^)	25.37	±	5.55	24.78	±	3.79	26.46	±	7.86	0.268
Sedentary Behaviour										
Sedentary bouts (bouts/day)	13.20	±	3.72	13.68	±	3.48	12.44	±	3.63	0.130
Sedentary time (min/day)	529.33	±	97.13	536.07	±	87.38	516.82	±	114.32	0.469
Sedentary breaks (breaks/day)	13.05	±	3.72	13.60	±	3.50	12.06	±	4.01	0.130
Sedentary breaks (min/day)	94.89	±	45.82	86.56	±	30.78	110.32	±	63.30	0.055
Maximum time in sedentary breaks (min/day)	137.58	±	62.35	127.38	±	35.24	156.42	±	92.38	0.085
Physical Activity										
PA expenditure (kcals/day)	315.29	±	124.60	325.05	±	128.56	297.18	±	117.78	0.413
LPA (min/day)	285.14	±	61.14	270.65	±	49.84	312.04	±	71.67	0.011
MPA (min/day)	45.59	±	16.65	43.13	±	13.25	48.56	±	21.68	0.315
VPA (min/day)	5.35	±	5.72	5.14	±	5.10	5.76	±	6.82	0.694
VVPA (min/day)	2.98	±	5.35	3.24	±	6.13	2.48	±	3.51	0.604
MVPA (min/day)	52.96		19.22	52.88		17.16	53.08		22.46	0.967
Oxidative Stress										
SOD (U/mg prot)	1.28	±	0.33	1.24	±	0.27	1.34	±	0.43	0.262
CAT (U/mg prot)	0.51	±	0.25	0.44	±	0.24	0.61	±	0.25	0.008
GPx (U/mg prot)	0.04	±	0.01	0.04	±	0.01	0.04	±	0.01	0.388
GSH (ng/mg prot)	18.78	±	11.01	16.67	±	5.20	22.68	±	16.79	0.043
MDA (µmol/L)	0.80	±	0.56	0.86	±	0.59	0.70	±	0.50	0.295
AOPPs (µmol/L)	311.72	±	138.88	345.09	±	138.89	249.73	±	118.49	0.010
Inflammation										
IL−6 (ng/mL)	0.48	±	0.35	0.56	±	0.37	0.33	±	0.26	0.026
TNF-α (ng/mL)	8.06	±	8.27	9.69	±	9.39	4.89	±	4.16	0.045

Values are expressed as mean ± standard deviation. Significant differences are highlighted in bold. PA, physical activity; LPA, light physical activity; MPA, moderate physical activity, VPA, vigorous physical activity; VVPA, very vigorous physical activity; MVPA, moderate-to-vigorous physical activity, SOD, superoxide dismutase; CAT, catalase; GPx, glutathione peroxidase; GSH, glutathione; MDA, malondialdehyde; AOPPs, advanced oxidation protein products; IL−6, interleukin−6; TNFα, tumour necrosis factor-α.

**Table 2 ijerph-20-00899-t002:** Associations of physical activity and sedentary behaviour with enzymatic antioxidant markers.

	SOD (U/mg prot)						CAT (U/mg prot)						GPx (U/mg prot)						GSH (ng/mg prot)					
	Men (*n* = 39)			Women (*n* = 21)			Men (*n* = 39)			Women (*n* = 21)			Men (*n* = 39)			Women (*n* = 21)			Men (*n* = 39)			Women (*n* = 21)		
	*β*	*R* ^2^	*p*	*β*	*R* ^2^	*p*	*β*	*R* ^2^	*p*	*β*	*R* ^2^	*p*	*β*	*R* ^2^	*p*	*β*	*R* ^2^	*p*	*β*	*R* ^2^	*p*	*β*	*R* ^2^	*p*
Sedentary Behaviour																								
Sedentary bouts (bouts/day)	0.149	0.022	0.354	−0.185	0.034	0.365	−0.122	0.015	0.448	−0.018	0.000	0.932	−0.050	0.002	0.758	−0.171	0.029	0.404	−0.025	0.001	0.876	−0.168	0.028	0.411
Sedentary time (min/day)	−0.016	0.000	0.918	−0.181	0.033	0.377	−0.315	0.099	0.045	−0.035	0.001	0.866	0.025	0.001	0.874	−0.017	0.000	0.933	−0.070	0.005	0.664	−0.014	0.000	0.947
Sedentary breaks (breaks/day)	0.149	0.022	0.354	−0.185	0.034	0.365	−0.122	0.015	0.448	−0.018	0.000	0.932	−0.050	0.002	0.758	−0.171	0.029	0.404	−0.025	0.001	0.876	−0.168	0.028	0.411
Sedentary breaks (min/day)	−0.135	0.018	0.401	0.074	0.006	0.718	−0.005	0.000	0.977	−0.009	0.000	0.967	0.009	0.000	0.955	−0.006	0.000	0.978	−0.046	0.002	0.774	0.006	0.000	0.977
Maximum time in sedentary breaks (min/day)	−0.236	0.055	0.138	−0.058	0.003	0.780	0.190	0.036	0.234	0.079	0.006	0.701	−0.046	0.002	0.775	−0.068	0.005	0.742	−0.097	0.009	0.546	−0.123	0.015	0.549
Physical Activity																								
PA expenditure (kcals/day)	−0.371	0.138	0.017	−0.081	0.007	0.693	0.066	0.004	0.683	−0.188	0.014	0.567	−0.086	0.007	0.593	0.066	0.004	0.749	−0.196	0.038	0.220	−0.038	0.001	0.855
LPA (min/day)	0.123	0.015	0.445	0.042	0.002	0.837	−0.121	0.015	0.452	−0.069	0.005	0.739	0.227	0.052	0.153	0.055	0.003	0.788	0.213	0.045	0.181	0.035	0.001	0.864
MPA (min/day)	−0.249	0.062	0.116	0.053	0.003	0.796	−0.210	0.044	0.187	−0.143	0.021	0.485	0.064	0.004	0.690	0.201	0.040	0.325	−0.180	0.032	0.260	0.184	0.034	0.369
VPA (min/day)	−0.142	0.020	0.377	0.091	0.008	0.659	−0.146	0.021	0.363	−0.130	0.017	0.526	−0.098	0.010	0.540	0.257	0.066	0.204	−0.136	0.018	0.397	0.186	0.035	0.361
VVPA (min/day)	−0.155	0.024	0.333	0.073	0.005	0.723	−0.018	0.000	0.913	−0.252	0.063	0.215	−0.017	0.000	0.916	0.120	0.014	0.560	0.120	0.014	0.456	−0.015	0.000	0.942
MVPA (min/day)	−0.285	0.081	0.071	0.088	0.008	0.670	−0.208	0.043	0.193	−0.212	0.045	0.299	0.014	0.000	0.932	0.283	0.080	0.161	−0.133	0.018	0.407	0.226	0.051	0.267

The symbol β means standardized coefficient. SOD, superoxide dismutase; CAT, catalase; GPx, glutathione peroxidase; GSH, glutathione; PA, physical activity; LPA, light physical activity; MPA, moderate physical activity, VPA, vigorous physical activity; VVPA, very vigorous physical activity; MVPA, moderate-to-vigorous physical activity.

**Table 3 ijerph-20-00899-t003:** Associations of physical activity and sedentary behaviour with plasma markers of oxidative stress and inflammation.

	MDA (µmol/L)						AOPPs (µmol/L)						IL-6 (ng/mL)						TNFα (ng/mL)					
	Men (*n* = 39)			Women (*n* = 21)			Men (*n* = 39)			Women (*n* = 21)			Men (*n* = 39)			Women (*n* = 21)			Men (*n* = 39)			Women (*n* = 21)		
	*β*	*R* ^2^	*p*	*β*	*R* ^2^	*p*	*β*	*R* ^2^	*p*	*β*	*R* ^2^	*p*	*β*	*R* ^2^	*p*	*β*	*R* ^2^	*p*	*β*	*R* ^2^	*p*	*β*	*R* ^2^	*p*
Sedentary Behaviour																								
Sedentary bouts (bouts/day)	0.187	0.035	0.255	−0.073	0.005	0.753	−0.156	0.024	0.330	−0.188	0.035	0.358	0.132	0.017	0.451	−0.092	0.008	0.717	0.061	0.004	0.683	0.024	0.001	0.902
Sedentary time (min/day)	0.205	0.042	0.211	−0.135	0.018	0.558	−0.077	0.006	0.633	−0.100	0.010	0.628	0.170	0.029	0.328	−0.019	0.000	0.942	−0.070	0.005	0.664	−0.014	0.000	0.947
Sedentary breaks (breaks/day)	0.187	0.035	0.255	−0.073	0.005	0.753	−0.156	0.024	0.330	−0.188	0.035	0.358	0.132	0.017	0.451	−0.092	0.008	0.717	0.061	0.004	0.683	0.024	0.001	0.902
Sedentary breaks (min/day)	−0.232	0.054	0.156	0.145	0.021	0.530	0.182	0.033	0.255	0.089	0.008	0.666	−0.063	0.004	0.718	−0.067	0.004	0.792	−0.199	0.039	0.181	−0.087	0.007	0.661
Maximum time in sedentary breaks (min/day)	**−0.308**	**0.095**	**0.049**	0.123	0.015	0.595	−0.004	0.000	0.982	−0.193	0.037	0.344	−0.070	0.005	0.691	−0.296	0.088	0.233	−0.108	0.012	0.470	−0.326	0.106	0.091
Physical Activity																								
PA expenditure (kcals/day)	**−0.348**	**0.121**	**0.030**	−0.091	0.008	0.695	−0.119	0.014	0.460	−0.198	0.039	0.332	0.257	0.066	0.137	−0.097	0.009	0.702	−0.019	0.000	0.897	−0.149	0.022	0.449
LPA (min/day)	0.263	0.069	0.106	0.091	0.008	0.695	−0.024	0.001	0.883	0.082	0.007	0.691	−0.152	0.023	0.383	0.014	0.000	0.957	0.060	0.004	0.687	0.019	0.000	0.925
MPA (min/day)	−0.143	0.020	0.387	0.163	0.027	0.480	0.154	0.024	0.335	0.124	0.015	0.545	0.116	0.013	0.508	0.049	0.002	0.846	0.094	0.009	0.531	−0.135	0.018	0.493
VPA (min/day)	0.205	0.042	0.211	−0.135	0.018	0.558	0.171	0.029	0.285	0.099	0.010	0.630	−0.062	0.004	0.724	−0.150	0.022	0.554	−0.002	0.000	0.991	**0.437**	**0.191**	**0.020**
VVPA (min/day)	−0.057	0.003	0.731	−0.284	0.081	0.212	−0.103	0.011	0.524	−0.198	0.039	0.332	**0.528**	**0.278**	**0.001**	0.110	0.012	0.664	0.014	0.000	0.923	0.142	0.020	0.471
MVPA (min/day)	−0.162	0.026	0.324	0.108	0.012	0.640	0.130	0.017	0.418	0.117	0.014	0.570	0.227	0.051	0.190	0.024	0.001	0.923	0.075	0.006	0.619	0.017	0.006	0.931

Significant associations are highlighted in bold. The symbol β means standardized coefficient. MDA, malondialdehyde; AOPPs, advanced oxidation protein products; IL-6, interleukin-6; TNFα, tumour necrosis factor-α; PA, physical activity; LPA, light physical activity; MPA, moderate physical activity, VPA, vigorous physical activity; VVPA, very vigorous physical activity; MVPA, moderate-to-vigorous physical activity.

## Data Availability

The data presented in this study are available on reasonable request from the corresponding author.

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
