# Peer review of "Sex-Specific Relationships of Physical Activity and Sedentary Behaviour with Oxidative Stress and Inflammatory Markers in Young Adults"

_ijerph, 2023, doi:10.3390/ijerph20020899_

Round 1

Reviewer 1 Report

General Comments:

This is a well-written manuscript of a study that aims to evaluate the relationships among objectively measured sedentary and PA behaviors and markers of inflammation and oxidative stress in men and women.  The primary finding was that oxidative stress markers were associated with PA and sedentary behavior but only in men.

Specific Comments:

1.       Line 71…it is recommended that additional details for the “NutAF research Project” be provided.  Have the results of this project been previously reported?  If so, provide the reference with additional details as to the study population.

2.       Line 82…How were the subjects recruited?  Advertisements, recruited on an academic campus, local community, etc.? 

3.       Line 94…It is stated that the back side of the hip placement allowed accurate estimation of sedentary behavior (as per provided reference), but does that placement alter the assessment of light, moderate, or vigorous physical activity counts/EE?

4.       Line 97…was sleep time captured and excluded from the analysis or were filters used in Actilife to exclude certain times of day (e.g., 10 PM to 6 AM) from the analyses?

5.       Lines 101-105…the Kozey-Keadle et al. reference provided only describes the cut-points used for time spent in sedentary activity.  Please provide a reference for the cut-points used for time spent in light, moderate, and vigorous activities.  Additionally, identify the methods used to determine PA energy expenditure.

6.       Line 122…Spell out the first use of the acronym TCA.

7.       Line 127…it is recommended that a reference be provided for the methods used to quantify MDA and AOPPs.

8.       Line 148…it is recommended that the authors provide additional comment on the strength of the significant relationships observed.  Most of the significant associations had r2 values in the range of 0.095-0.138.  How does the low “goodness-of-fit” for the models impact the findings?

9.       Line 244…it is recommended that the small sample size, particularly for the women, be highlighted as a limitation when using regression analyses, especially for variables that are likely impacted by measurement error as can be the case with PA assessment with accelerometers.

Author Response

Response to Reviewer 1 Comments

This is a well-written manuscript of a study that aims to evaluate the relationships among objectively measured sedentary and PA behaviors and markers of inflammation and oxidative stress in men and women. The primary finding was that oxidative stress markers were associated with PA and sedentary behavior but only in men.

Point 1: Line 71…it is recommended that additional details for the “NutAF research Project” be provided.  Have the results of this project been previously reported?  If so, provide the reference with additional details as to the study population.

Response 1: Thank you very much for your comments and valuable review. Regarding to Point 1, yes, it does. The project has some previous publications, the cites/references have been included in this section, as well as the general aim of the NutAF project as follows: “which aimed to investigate the relationships between nutritional habits and the level of physical activity with appetite regulating hormones and genetic polymorphisms related to obesity in young adults [new CITES]”.

Point 2: Line 82…How were the subjects recruited?  Advertisements, recruited on an academic campus, local community, etc.?

Response 2: The participant recruitment has been described as follows: “Participant recruitment consisted of mailed flyers describing the research and contact information, the flyers were also published on social media (Facebook, Instagram, etc.), printed flyers were posted on academic campus (at bus stops, housing complexes, and near dorms) and around the community targeted young adult high traffic areas”. We appreciate this comment since it is a relevant information.

Point 3: Line 94…It is stated that the back side of the hip placement allowed accurate estimation of sedentary behavior (as per provided reference), but does that placement alter the assessment of light, moderate, or vigorous physical activity counts/EE?

Response 3: Yes, it is. Back side of the hip placement and wrist placement provide accurate estimation of light, moderate, or vigorous physical activity counts/EE; however, only back side of the hip placement is able to describe sedentary behaviour while wrist placement must indicate inactivity (because it is not able to detect the sedentary position according to the position of the accelerometer axes). To avoid possible misunderstandings, we have also included that this placement is suitable for estimating the PA levels since our sentence was not properly written in our first submission.

Point 4: Line 97…was sleep time captured and excluded from the analysis or were filters used in Actilife to exclude certain times of day (e.g., 10 PM to 6 AM) from the analyses?

Response 4: Yes, it was. We have included that “Participants were asked to remove the accelerometer during hours of sleep”. Thank you.

Point 5: Lines 101-105…the Kozey-Keadle et al. reference provided only describes the cut-points used for time spent in sedentary activity.  Please provide a reference for the cut-points used for time spent in light, moderate, and vigorous activities.  Additionally, identify the methods used to determine PA energy expenditure.

Response 5: We agree with the reviewer. We have corrected this section with the following reference regarding PA (https://doi.org/10.1016/j.jsams.2011.04.003) and the Kozey-Keadle et al. reference following the sedentary behaviour information.

Point 6: Line 122…Spell out the first use of the acronym TCA.

Response 6: Corrected after suggestion. TCA is trichloroacetic acid.

Point 7: Line 127…it is recommended that a reference be provided for the methods used to quantify MDA and AOPPs.

Response 7: We agree with the reviewer. Both references have been included after suggestion.

Point 8: Line 148…it is recommended that the authors provide additional comment on the strength of the significant relationships observed.  Most of the significant associations had r2 values in the range of 0.095-0.138.  How does the low “goodness-of-fit” for the models impact the findings?

Response 8: Although these associations could seem quite low as the reviewer highlights, which indicates a relationship but with a high variability related to the distribution of data points around the fitted line, it is important to note that we are included oxidative stress markers and PA levels and sedentary behaviour. Despite the fact that these outcomes can be related as it is supporting by our findings and it has been discussed, many other variables can influence them, human behaviour is complex. Therefore, we have include this aspect as a limitation according to the reviewer’s comment (see Point 9).

Point 9: Line 244…it is recommended that the small sample size, particularly for the women, be highlighted as a limitation when using regression analyses, especially for variables that are likely impacted by measurement error as can be the case with PA assessment with accelerometers.

Response 9: It has been included as a limitation after the reviewer’s suggestion. Thus, point 8 and 9 have been addressed as follows: “… Secondly, studies with a higher sample size, particularly for the women which was small in the present study, are encouraged. Finally, it is important to highlight the low strength of relationships reported in most of the significant associations (low R-squared) indicating a high variability, since other relevant factor could influence these outcomes”.

Reviewer 2 Report

The authors look at various inflammatory markers and oxidative stress on young adults as related to the physical activity.

Comments

The results section needs to be substantially revised. The results presented in table 1, 2 and 3 need to be described appropriately in the text before an evaluation of the paper can be made. It is not enough to present tables without description of the key findings in words.

Additionally it should be discussed/ addressed what were the baseline values of all the indices measured. It is not clear if the baseline values are in table 1.

Also when did blood sampling occurred? At the end only, at the beginning or both? It is not clear. Please comment

Given that the blood sampling for measurements took place at the end, it would be important to know if the days before the study started had same/normal daily physical activity, or for any reason they had activities that would skew the results of the study as a result. 

Author Response

The authors look at various inflammatory markers and oxidative stress on young adults as related to the physical activity.

Point 1: The results section needs to be substantially revised. The results presented in table 1, 2 and 3 need to be described appropriately in the text before an evaluation of the paper can be made. It is not enough to present tables without description of the key findings in words.

Response 1: Thank you very much for your comments and valuable review. We have detailed in the text the main results that are presented in tables in the hope that it is better written in this submission. All changes have been highlighted in yellow.

Point 2: Additionally it should be discussed/ addressed what were the baseline values of all the indices measured. It is not clear if the baseline values are in table 1.

Response 2: Fasting blood samples were taken from the antecubital vein in the morning only once. We are reporting baseline values in a cross-sectional study investigating the relationships of oxidative stress variables and PA levels and sedentarism. Blood samples were collected only in one day for each participant. Therefore, table 1 provides the Mean and Standard deviation in all the sample and dividing by sex.

Point 3: Also when did blood sampling occurred? At the end only, at the beginning or both? It is not clear. Please comment

Response 3: Only at the end: “The physical activity levels and sedentary behaviour were estimated using accelerometers for seven consecutive days immediately before the day of blood sample collection” … “After returning the accelerometer, fasting blood samples were taken from the antecubital vein”. This has been specified in the Methods section. Thank you.

Point 4: Given that the blood sampling for measurements took place at the end, it would be important to know if the days before the study started had same/normal daily physical activity, or for any reason they had activities that would skew the results of the study as a result.

Response 4: They were asked to maintain same/normal daily physical activity along the week except the day before measurements in which they should avoid vigorous physical exercise. It has been detailed in the Methods “Participants were asked to maintain normal daily physical activity along the week, except the day before measurements in which they should avoid vigorous physical exercise. Also, the morning of blood sampling, the participants avoided active transport (walking, cycling, etc.)”. We really appreciate the reviewer’s comments and time.

Round 2

Reviewer 1 Report

Thank you for addressing the previous comments.  Please consider the following revisions to improve the methodology utilized to analyze the PA data:

1.  Lines 110-111...Delete "All the accelerometer data were analysed using ActiLife 6.6.2 software (ActiGraph, Florida, USA).

2.  For the paragraph starting  with line 112 revise to read "Accelerometer data were analysed using ActiLife 6.6.2 software (ActiGraph, Florida, USA). The Freedson Adult VM3 (2011) cut points were utilized to classify physical activity into light (LPA, 150–2689 counts/min), moderate (MPA, 2690–6166 counts/min), vigorous (VPA, 6167-9642 counts/min), and very vigorous (VVPA, ≥9643 counts/min) physical activity [24].  Moderate to vigorous physical activity (MVPA) was considered any activity from moderate to very vigorous (i.e. ≥2690 counts/min).  Additionally, activity energy expenditure was estimated using the Freedson VM3 (2011) equation.  Sedentary behaviour (classified as <150 counts/min), sedentary bouts as the time ac-cumulated in a consecutive period of >10 min, the mean number of sedentary bouts, and the time spent in sedentary bouts were calculated per day. Also, sedentary breaks were considered whenever participants were above 150 counts/min after a sedentary bout, and the time, number, and length of sedentary breaks were calculated per day [25].

3.  Line 262...would recommend replacing "There are some limitations to making solid conclusions from this study" to "This study has several limitations."

Author Response

Response to Reviewer 1 Comments

Thank you for addressing the previous comments.  Please consider the following revisions to improve the methodology utilized to analyze the PA data:

Point 1: Lines 110-111...Delete "All the accelerometer data were analysed using ActiLife 6.6.2 software (ActiGraph, Florida, USA).

Response 1: Done.

Point 2: For the paragraph starting  with line 112 revise to read "Accelerometer data were analysed using ActiLife 6.6.2 software (ActiGraph, Florida, USA). The Freedson Adult VM3 (2011) cut points were utilized to classify physical activity into light (LPA, 150–2689 counts/min), moderate (MPA, 2690–6166 counts/min), vigorous (VPA, 6167-9642 counts/min), and very vigorous (VVPA, ≥9643 counts/min) physical activity [24].  Moderate to vigorous physical activity (MVPA) was considered any activity from moderate to very vigorous (i.e. ≥2690 counts/min).  Additionally, activity energy expenditure was estimated using the Freedson VM3 (2011) equation.  Sedentary behaviour (classified as <150 counts/min), sedentary bouts as the time ac-cumulated in a consecutive period of >10 min, the mean number of sedentary bouts, and the time spent in sedentary bouts were calculated per day. Also, sedentary breaks were considered whenever participants were above 150 counts/min after a sedentary bout, and the time, number, and length of sedentary breaks were calculated per day [25].

Response 2: Done. Thank you very much.

Point 3: Line 262...would recommend replacing "There are some limitations to making solid conclusions from this study" to "This study has several limitations."

Response 3: Corrected after suggestion.

Reviewer 2 Report

Previous comments have been adressed.

One minor addtion point: Although the concussions are supported by the data regarding the protection women have from oxidative stress, it would be nice to briefly state what future studies are needed  (future directions) to make this finding strogner. 

Author Response

Response to Reviewer 2 Comments

Previous comments have been adressed.

Point 1: Although the concussions are supported by the data regarding the protection women have from oxidative stress, it would be nice to briefly state what future studies are needed  (future directions) to make this finding strogner.

Response 1: It has been included after suggestion as follows “Future studies with an experimental design are encouraged in order to determine the impact of different exercise programs on oxidative stress and inflammatory markers according to sex, ideally including not only blood samples but muscle biopsies as well”. Indeed, we have recently obtained funded to carry out this project and we are investigating the effect of a 12-week exercise program, comparing HIIT vs. MICT, on oxidative stress, inflammation, and mitochondrial respiration in sedentary men and women. Thank you very much.
